# LPS-Induced Garcia Effect and Its Pharmacological Regulation Mediated by Acetylsalicylic Acid: Behavioral and Transcriptional Evidence

**DOI:** 10.3390/biology12081100

**Published:** 2023-08-07

**Authors:** Veronica Rivi, Anuradha Batabyal, Ken Lukowiak, Cristina Benatti, Giovanna Rigillo, Fabio Tascedda, Joan M. C. Blom

**Affiliations:** 1Department of Biomedical, Metabolic and Neural Sciences, University of Modena and Reggio Emilia, 41125 Modena, Italy; cristina.benatti@unimore.it (C.B.); giovanna.rigillo@unimore.it (G.R.); 2Department of Physiology and Pharmacology, Hotchkiss Brain Institute, University of Calgary, Calgary, AB T2N 4N1, Canada; anuradha.batabyal@ucalgary.ca (A.B.); lukowiak@ucalgary.ca (K.L.); 3Department of Physical and Natural Sciences, FLAME University, Pune 412115, Maharashtra, India; 4Centre of Neuroscience and Neurotechnology, University of Modena and Reggio Emilia, 41125 Modena, Italy; fabio.tascedda@unimore.it; 5Department of Life Sciences, University of Modena and Reggio Emilia, 41125 Modena, Italy; 6CIB, Consorzio Interuniversitario Biotecnologie, 34148 Trieste, Italy

**Keywords:** inflammation, learning, memory, glutamate receptors, *HSP70*, *TRL4*, *CREB1*

## Abstract

**Simple Summary:**

Using our *Lymnaea stagnalis* model systems and combining that with a Garcia effect training procedure, we studied novel aspects of this complex and highly conserved conditioned behavior and its pharmacological regulation. Injecting snails with lipopolysaccharide (LPS) 25 μg 1 h after snails experienced a novel taste caused snails to form a long-lasting Garcia-effect memory to avoid that specific taste. This effect was prevented by the pre-exposure of snails to acetylsalicylic acid (ASA) for 1 h before the LPS injection. Here, we researched the transcriptional effects of ASA and LPS in the snails’ central nervous system alone and in combination with naive snails. In a similar manner, the behavioral and molecular mechanisms causing the LPS-induced Garcia effect and its mitigation by ASA were studied. The LPS injections enhanced the expression levels of immune and stress response targets and enhancement was prevented by pre-exposure to ASA. Regarding genes associated with neuroplasticity, LPS by itself did not affect their expression levels. However, when combined with the Garcia-effect training procedure they were upregulated consistent with LTM formation. These findings suggest a conserved crosstalk between the immune and central nervous systems.

**Abstract:**

*Lymnaea stagnalis* learns and remembers to avoid certain foods when their ingestion is followed by sickness. This rapid, taste-specific, and long-lasting aversion—known as the Garcia effect—can be formed by exposing snails to a novel taste and 1 h later injecting them with lipopolysaccharide (LPS). However, the exposure of snails to acetylsalicylic acid (ASA) for 1 h before the LPS injection, prevents both the LPS-induced sickness state and the Garcia effect. Here, we investigated novel aspects of this unique form of conditioned taste aversion and its pharmacological regulation. We first explored the transcriptional effects in the snails’ central nervous system induced by the injection with LPS (25 mg), the exposure to ASA (900 nM), as well as their combined presentation in untrained snails. Then, we investigated the behavioral and molecular mechanisms underlying the LPS-induced Garcia effect and its pharmacological regulation by ASA. LPS injection, both alone and during the Garcia effect procedure, upregulated the expression levels of immune- and stress-related targets. This upregulation was prevented by pre-exposure to ASA. While LPS alone did not affect the expression levels of neuroplasticity genes, its combination with the conditioning procedure resulted in their significant upregulation and memory formation for the Garcia effect.

## 1. Introduction

Using our model system, the pond snail *Lymnaea stagnalis*, we have been able to begin to uncover aspects of the causal neuronal mechanisms underlying learning and its subsequent consolidation into long-term memory (LTM) [1,2,3,4,5,6,7,8,9,10,11,12,13,14,15,16,17,18,19,20].

We recently demonstrated—for the first time in an invertebrate model organism—that *Lymnaea* is capable of forming a unique type of conditioned taste aversion known as the “Garcia effect” [21,22,23,24,25,26]. To induce a Garcia effect, a single pairing of a novel appetitive stimulus followed by nausea or sickness even hours later is sufficient to establish an LTM that causes a long-lasting avoidance of that food [24,25,27].

In our previous studies, we found that an injection of lipopolysaccharide (LPS), an activator of the immune system, can be used as a sickness-inducing stimulus to cause a Garcia effect lasting for at least 24 h [27]. Interestingly, we also demonstrated that the administration of a non-steroidal anti-inflammatory drug, acetylsalicylic acid (ASA), prior to the LPS injection, prevented both the induction of the sickness state caused by LPS [28] and the occurrence of the Garcia effect [27]. These findings have opened new avenues for research, seeking to elucidate the molecular mechanisms that underlie the Garcia effect and provide a comprehensive understanding of the intricate interplay between the immune and central nervous systems.

In this study, we aimed to investigate the transcriptional effects of ASA and LPS exposure alone and their combined presentation in the central ring ganglia of untrained snails. Additionally, we aimed to unravel the behavioral and molecular mechanisms involved in the LPS-induced Garcia effect and how ASA mediates its pharmacological regulation.

To achieve this, we examined the transcriptional changes of specific targets associated with the regulation of immune and stress responses:-the *Toll-like Receptor 4* (*LymTLR4*), a key component for the innate immunity of invertebrates [29,30] and in mammals, mediates LPS-induced immune response [31];-the *Molluscan Defense Molecule* (*LymMDM*), an Ig-superfamily member, which allows mollusks to mount an effective immune response and ensure their survival [32],-the *Heat Shock Protein 70* (*LymHSP70*), which plays a key conserved role in stress response [33,34,35].

We hypothesized that (1) the injection of LPS would lead to an immune and stress response in snails, characterized by an upregulation of *LymTLR4*, *LymMDM*, and *LymHSP70* mRNA levels, and (2) this upregulation would be prevented by exposure to ASA. Moreover, we hypothesized that the formation of the Garcia effect would also affect the mRNA levels of ionotropic glutamatergic receptors and the transcription factor cAMP response element-binding protein 1 (*LymCREB1*), known to be involved in memory consolidation [36,37,38,39].

Thus, we paid particular attention to the glutamate ionotropic receptor NMDA type subunit 1 (*LymGRIN1*), 2A (*LymGRIN2A*), and 2B (*LymGRIN2B*), the glutamate ionotropic receptor AMPA type subunit 1 (*LymGRIA1*), as well as *LymCREB1*, because of their involvement in mediating neuroplasticity, including learning and memory [36,37,40,41,42,43,44].

Thus, to test our overall hypothesis, this study was organized as follows:

In Experiment 1, we centered our attention on the transcriptional effects on the abovementioned targets induced by (1) LPS, the sickness-inducing stimulus used in the Garcia effect training procedure, (2) ASA, the anti-inflammatory drug used to prevent the LPS effect, and (3) the ASA exposure before the LPS injection, in untrained snails.

In our second experiment, we studied the effects induced by LPS injection, ASA exposure, and ASA exposure before LPS injection on the ability of snails to form a Garcia effect.

In Experiment 3, we studied the transcriptional effects elicited by the Garcia effect training and its prevention mediated by ASA on the expression levels of target genes.

## 2. Materials and Methods

### 2.1. Animals

The freshly collected ‘Margo snails’ used in this study, were collected from Margo Lake in Saskatoon, Saskatchewan, with coordinates of 51°49′ N and 103°21′1.8″ W, and an elevation of 526 m. Adult snails, measuring 2.5–3.0 cm in shell length, were housed in well-oxygenated artificial pond water at a density of one snail per liter of water. The artificial pond water was prepared by mixing deionized water with 26 g/L of Instant Ocean (Spectrum Brands, Madison, WI, USA). To maintain a standard calcium level of 80 mg/L, calcium sulfate dihydrate was added to the water.

The snails were kept in a room with a temperature ranging from 20 to 22 °C, following a 16 h light and 8 h dark cycle. They were provided with unrestricted access to romaine lettuce for feeding purposes.

### 2.2. LPS Treatment

Margo snails were injected with 25 μg of Escherichia coli-derived lipopolysaccharide (LPS) serotype O127:B8 (L3129), which is approximately equivalent to 8 mg/kg. The LPS solution was prepared by dissolving 625 μg of LPS in 1 mL of snail saline solution, composed of 51.3 mM NaCl, 1.7 mM KCl, 5.0 mM MgCl_2_, 1.5 mM CaCl_2_, and 5.0 mM HEPES, with a pH of 7.9–8. A volume of 40 μL of the LPS solution was injected into the abdominal body cavity of each snail. Snails in the control group were sham-injected with 40 μL of snail saline. Following the injection, the snails were kept in an upside-down position without being immersed in artificial pond water for 10 min, as described in previous studies [27,45].

### 2.3. Acetylsalicylic Acid (ASA) Treatment

In this study, we bought locally grocery store-purchased acetylsalicylic acid (Bayer, Leverkusen, Germany) tablets with a purity of at least 99.0%. Through pilot experiments, we determined that a concentration of 900 nM of ASA would not affect important homeostatic behaviors such as aerial respiration and feeding [28]. To prepare the ASA solution, we dissolved one tablet (81 mg, molecular weight: 180.158 g/mol) in 500 mL of artificial pond water. In this study, snails assigned to the ASA treatment group were placed in a 1 L beaker containing 500 mL of ASA-artificial pond water and kept there for 1 h.

### 2.4. Behavioral Procedure to Induce a Garcia Effect

Rasping behavior in *Lymnaea* refers to a rhythmic motor activity where the snails repeatedly scrape their radulae against a substrate, allowing them to consume food [46,47,48]. After being acclimated in the carrot slurry for 3 min, the number of rasps elicited by the carrot slurry was recorded over a period of 2 min. Following this initial observation, the snails were returned to their aquaria for 1 h before being injected with either LPS or snail saline solution. After 3 h from the injection, the rasping behavior elicited by the carrot slurry was again recorded for 2 min, preceded by a 3 min acclimation period. To prepare the carrot slurry, we blended two organic carrots and strained them together with 500 mL of artificial pond water. The snails were placed in a Petri dish with a diameter of 14 cm, and the dish was filled with enough carrot slurry to partially submerge the snails. For better visibility of the rasping behavior, the Petri dishes were positioned on a clear Plexiglas stand elevated 10 cm above a mirror. The average number of rasps per minute induced by carrot slurry was determined to be 19.47 ± 3.2 (mean ± SEM) based on sample size (N) of 15 Margo snails.

### 2.5. Experimental Design

#### 2.5.1. Experiment 1: Transcriptional Effects of ASA, LPS, and Their Paired Exposure

In this experiment, we used 4 naïve cohorts of snails (N = 8 for each cohort): (1) Snails of the ‘Saline group’ were injected with snail saline; (2) snails of the ‘LPS group’ were injected with LPS; (3) snails of the ‘ASA group’ experienced ASA for 1 h; and (4) snails of the ‘ASA_LPS group’ experienced ASA for 1 h before being injected with LPS. Three hours after the treatments, snails were euthanized in ice for 10 min, and the central ring ganglia were dissected. These doses of LPS (25 mg) and ASA (900 nM) have been successfully adopted in previous studies [27,28]. The central ring ganglia were stored at −80° before before analysis.

#### 2.5.2. Experiment 2: LPS-Induced Garcia Effect Procedure and Its Pharmacological Regulation Mediated by ASA (Behavioral Data)

In This Study, 8 Groups of Naïve Pond-Collected Snails (N = 8, Each Group) Were Used.

Snails of the ‘Saline_C group’ were first injected with snail saline, and 3 h later, were exposed to the novel food (carrot slurry) for 2 min, and the number of rasps was recorded. Thus, these snails were not exposed to the novel taste (i.e., carrot slurry) before the injection.Snails of the ‘Saline group’ were exposed to carrot slurry for 2 min and the number of rasps was counted. One hour later, snails were injected with snail saline. After 3 h, snails were re-exposed to carrot slurry for 2 min and the number of rasps elicited by the carrot slurry was again recorded.Snails of the ‘LPS-C group’ were injected with LPS and 3 h later were exposed to the carrot slurry for the first time. The number of rasps elicited by the carrot slurry was again recorded for 2 min.Snails of the ‘LPS group’ were exposed to carrot slurry for 2 min, during which the number of rasps was counted. One hour later, snails were injected with LPS and, 3 h later, were re-exposed to carrot slurry for 2 min, and the number of rasps elicited by the carrot slurry was again recorded.Snails of the ‘ASA group’ were exposed to carrot slurry for 2 min and the number of rasps elicited by the carrot slurry was counted. One hour later, snails were exposed to ASA for 1 h. Three hours later, snails were re-exposed to carrot slurry for 2 min and the number of rasps elicited by the carrot slurry was again recorded.Snails of the ‘ASA_Sal group’ were exposed to carrot slurry for 2 min and the number of rasps was counted. One hour later, snails were exposed to ASA for 1 h and immediately after were injected with snail saline. After 3 h, snails were re-exposed to carrot slurry for 2 min and the number of rasps elicited by the carrot slurry was again recorded.Snails of the ‘Garcia effect group’ were exposed to carrot slurry for 2 min and the number of rasps elicited by the carrot slurry was counted. One hour later, snails were injected with LPS. Three hours later, snails were re-exposed to carrot slurry for 2 min and the number of rasps elicited by the carrot slurry was again recorded.Snails of the ‘ASA_LPS group’ were exposed to carrot slurry for 2 min and the number of rasps elicited by the carrot slurry was counted. One hour later, snails were exposed to ASA for 1 h and immediately after were injected with LPS. Three hours later, snails were re-exposed to carrot slurry for 2 min and the number of rasps elicited by the carrot slurry was again recorded.

#### 2.5.3. Experiment 3: Transcriptional Effects Induced by the LPS-Induced Garcia Effect and Its Pharmacological Regulation Mediated by ASA

Immediately after the memory test (Experiment 2), snails were euthanized in ice for 10 min, and the central ring ganglia were dissected. First, we performed control experiments to compare the mRNA levels of the selected targets between (1) snails of the ‘Saline group’ and those of the ‘Saline_C group’, and between (2) snails of the ‘ASA group’ and those of the ‘ASA_Saline group’. Then, we investigated the transcriptional effects induced by the LPS-induced Garcia effect and its pharmacological regulation mediated by ASA by comparing the mRNA levels of the selected targets between snails of the ‘Saline group’, ‘ASA_Saline group’, ‘LPS group’, ‘Garcia effect group’, and ‘ASA_LPS group’.

### 2.6. Total RNA Extraction, Reverse Transcription, and Real-Time Quantitative PCR

Total RNA extraction and DNAse treatment were conducted following previously described protocols [15]. Real-time quantitative PCR (RT-qPCR) was performed using 20 ng of mRNA, as previously described [15]. Specific forward and reverse primers were used at the final concentration of 300 nM (Table 1). As this is the first time in which the mRNA levels of GRIN2A, GRIN2B, and GRIA1 have been measured and compared in a *Lymnaea* study, interested readers can find the detailed description of the identification and characterization of the transcripts of LymGRIN2A, LymGRIN2B, and LymGRIA in Appendix A. The mRNA levels of each target gene were normalized to the arithmetic mean between two housekeeping genes, elongation factor 1α (EF1α) and b-tubulin (βTUB).

No significant alterations were observed in the mRNA levels of the reference genes across the experimental procedures (one-way analysis of variance [ANOVA]), and the amplification efficiency of both the target genes and the reference genes was similar. For the quantitative evaluation of changes, the comparative 2−ΔΔCt method was performed using as a calibrator the average levels of expression of control animals (i.e., saline-injected snails in Experiment 1 and snails exposed to carrot 1 h before e 3 h after the saline injection).

### 2.7. Data Analysis

In the molecular experiments, we assessed the normality of our data using the Kolmogorov–Smirnov one-sample test for normality, considering the K-S distance and associated *p*-values. The analysis revealed that all targets exhibited a normal distribution. To compare the expression levels of each target between Experiments 1 and 3, we employed one-way ANOVA. To identify significant differences, Tukey’s post hoc test was applied. For the behavioral data analysis (Experiment 2), a paired Student’s *t*-test was used to compare the number of rasps elicited by carrot slurry before and 3 h after the treatment. The number of rasps elicited by carrot slurry between snails of the Saline_C group and control naïve snails as well as the number of rasps elicited by carrot slurry between snails of the LPS group and control naïve snails was compared using an unpaired *t*-test. All tests were defined as significant at *p* < 0.05. Data were presented as mean ± standard error (SEM). Statistical analyses were conducted using IBM SPSS Statistics version 26.0 (IBM Corp., Armonk, NY, USA). Graphs were created using GraphPad Prism version 9.5.1e for Microsoft^®^ (GraphPad Software, Inc., La Jolla, CA, USA).

## 3. Results

### 3.1. Experiment 1: Transcriptional Effects of ASA, LPS, and Their Paired Exposure

The aim of Experiment 1 was to answer the following question: what are the transcriptional effects induced by ASA, LPS, and their paired presentation in the central ring ganglia of *Lymnaea*? To answer this question, we investigated whether the different ASA exposure for 1 h, the LPS injection, and the exposure to ASA for 1 h before the LPS injection would affect the expression levels of selected target genes involved in immune and stress response, or neuroplasticity. A main effect of the treatments was observed for *LymTLR4* [F (3, 28) = 14.74, R^2^ = 0.61, *p* < 0.001], *LymMDM* [F (3, 28) = 6.85, R^2^ = 0.42, *p* = 0.0013], and *LymHSP70* [F (3, 28) = 6.26, R^2^ = 0.41, *p* = 0.002] (Figure 1A–C). Tukey’s multiple comparisons tests showed significant upregulation of the expression levels of these targets in LPS-injected snails compared to the other groups (*LymTLR4*: LPS vs. Sal: *p* = 0.006, q = 6.4; vs. ASA: *p* < 0.0001, q = 8.87; vs. ASA_LPS: *p* = 0.002, q = 6.82; *LymMDM*: LPS vs. Sal: *p* = 0.002, q = 5.6; vs. ASA: *p* = 0.01, q = 4.7; vs. ASA_LPS: *p* = 0.005, q = 5.2; *LymHSP70*: LPS vs. Sal: *p* = 0.01, q = 5.6; vs. ASA: *p* = 0.003, q = 5.4; vs. ASA_LPS: *p* = 0.01, q = 4.8). No main effects of the treatment were observed for neuroplasticity targets: *LymGRIN1* [F (3, 28) = 0.97, R^2^ = 0.09, *p* = 0.42], *LymGRIN2A* [F (3, 28) = 0.65, R^2^ = 0.05, *p* = 0.55], *LymGRIN2B* [F (3, 28) = 1.83, R^2^ = 0.16, *p* = 0.16], *LymGRIA1* [F (3, 28) = 0.79, R^2^ = 0.07, *p* = 0.51], and *LymCREB1* [F (3, 28) = 0.18, R^2^ = 0.02, *p* = 0.91] (Figure 1D–H).

### 3.2. Experiment 2: LPS-Induced Garcia Effect Procedure and Its Pharmacological Regulation Mediated by ASA

Previously, we showed that the administration of the anti-inflammatory drug ASA before LPS injection can prevent the LPS-induced Garcia effect in Margo snails [27]. To further investigate this phenomenon at both the behavioral and molecular levels, we conducted Experiment 2, which involved various experimental conditions.

First, we confirmed that injecting snails with snail saline (i.e., Saline_C control group) did not affect feeding behavior elicited by carrot slurry, a novel appetitive stimulus. The number of rasps elicited by the carrot slurry in the Saline_C group was not significantly different from that in control Margo snails (unpaired *t*-test: *t* = 0.62, df = 21, *p* = 0.54) (Figure 2A). Next, a sham-injected control group (i.e., Snail saline group) was exposed to carrot slurry, and the feeding response was recorded. These snails were injected with snail saline 1 h later, and feeding behavior in response to carrot was evaluated again 3 h post-injection. Feeding behavior before and after the saline injection was not significantly different (paired *t*-test: *t* = 0.52, df = 6, *p* = 0.62). This confirms previous findings that the combination of a novel taste and the injection itself does not induce a Garcia effect (Figure 2B). Moreover, we injected LPS into a group of naïve snails (i.e., LPS group) who had not been exposed to the carrot slurry before. The rasping activity in response to the carrot slurry was recorded 3 h after the LPS injection. Comparing the LPS group to control naïve Margo snails, we found no significant differences in the response to the taste (unpaired *t*-test: *t* = 0.82, df = 21, *p* = 0.42) (Figure 2C). To assess the formation of the Garcia effect, a naive group of snails (i.e., Garcia effect group) was exposed to the novel carrot slurry and, one hour later, injected with LPS. As shown in Figure 2D, at 3 h post-injection, the rasping behavior in response to the carrot slurry was significantly decreased compared to the initial exposure (*t* = 7.54, df = 7, *p* = 0.0001), indicating the formation of the Garcia effect.

In a naïve group of snails (i.e., ASA group), the number of rasps elicited by the carrot slurry was recorded 1 h before and 3 h after a 1 h exposure to ASA. As reported in Figure 2E, there were no significant differences in the feeding response before and after the ASA exposure (*t* = 0.57, df = 6, *p* = 0.58), indicating that ASA did not significantly alter the positive hedonic effect of the carrot slurry. Consistent with our previous studies [27], we found that the combined exposure to ASA and saline injection did not result in a reduced carrot-induced feeding response. Indeed, snails of the ASA-Saline group were exposed to the carrot slurry for 2 min, and their feeding response was recorded. They experienced ASA for 1 h and then immediately injected with snail saline. Their feeding response elicited by carrot 3 h post-injection was not significantly altered by the combined treatment (*t* = 0.12, df = 7, *p* = 0.12) (Figure 2F). Finally, we investigated the prophylactic effect of ASA on the LPS-induced Garcia effect. Thus, snails in the ASA_LPS group were exposed to the novel carrot slurry and, one hour later, exposed to ASA for 1 h immediately before being injected with LPS. The number of rasps was then counted at 3 h post-injection. No significant differences were observed between the number of rasps in the carrot slurry before injection and at 3 h post-injection (*t* = 1.37, df = 7, *p* = 0.22). That is, the ASA exposure effectively prevented the LPS-induced Garcia effect (Figure 2G). It is important to note that this experimental procedure introduced a 2 h time gap between LPS injection and the initial exposure to the carrot slurry, whereas previous experiments had a 1 h gap. However, we have already shown that an LPS injection 2 h after the initial carrot slurry exposure also leads to feeding suppression and the formation of a Garcia effect in snails [27,28].

### 3.3. Experiment 3: Transcriptional Effects Induced by the LPS-Induced Garcia Effect and Its Pharmacological Regulation Mediated by ASA

The aim of Experiment 3 was to investigate the transcriptional effects induced by the LPS-induced Garcia effect and its pharmacological regulation mediated by ASA on the expression levels of selected targets involved in immune and stress response, or neuroplasticity. Therefore, immediately after the behavioral procedures, the central ring ganglia of the snails were dissected, and the RNA was extracted and reverse-transcribed to assess mRNA expression levels of *LymTLR4*, *LymMDM*, *LymHSP70*, *LymGRIN1*, *LymGRIN2A*, *LymGRIN2B*, *LymGRIA1*, and *LymCREB1* (Figure 3). First, we performed control experiments to compare the mRNA levels of the selected targets between snails of the ‘Saline group’ and those of the ‘Saline_C group’ and Snails of the ‘ASA group’ and those of the ‘ASA_Saline group’. No significant differences were found in the expression levels of *LymTLR4*, *LymMDM*, *LymHSP70*, *LymGRIN1*, *LymGRIN2A*, *LymGRIN2B*, *LymGRIA1*, and *LymCREB1* between snails of the ‘Saline group’ and those of the ‘Saline_C group’ (unpaired *t*-test: *LymTLR4*: *t* = 0.33, df = 14, *p* = 0.75; *LymMDM*: *t* = 0.94, df = 14, *p* = 0.36; *LymHSP70*: *t* = 0.094, df = 14, *p* = 0.96; *LymGRIN1*: *t* = 0.016, df = 14, *p* = 0.99; *LymGRIN2A*: *t* = 0.29, df = 14, *p* = 0.97; *LymGRIN2B*: *t* = 0.28, df = 14, *p* = 0.78; *LymGRIA1*: *t* = 0.48, df = 14, *p* = 0.64; *LymCREB1*: *t* = 0.24, df = 14, *p* = 0.81) (Appendix A).

Similarly, no significant differences were found in the expression levels of *LymTLR4*, *LymMDM*, *LymHSP70*, *LymGRIN1*, *LymGRIN2A*, *LymGRIN2B*, *LymGRIA1*, and *LymCREB1* between snails of the ‘ASA group’ and those of the ‘ASA_Sal group’ (unpaired *t*-test: *LymTLR4*: *t* = 0.59, df = 14, *p* = 0.56; *LymMDM*: *t* = 0.49, df = 14, *p* = 0.63; *LymHSP70*: *t* = 0.68, df = 14, *p* = 0.51; *LymGRIN1*: *t* = 1.13, df = 14, *p* = 0.28; *LymGRIN2A*: *t* = 1.59, df = 14, *p* = 0.14; *LymGRIN2B*: *t* = 0.76, df = 14, *p* = 0.46; *LymGRIA1*: *t* = 0.08, df = 14, *p* = 0.93; *LymCREB1*: *t* = 0.28, df = 14, *p* = 0.79) (Appendix A). Then, we investigated the transcriptional effects induced by the LPS-induced Garcia effect and its pharmacological regulation mediated by ASA by comparing the mRNA levels of the selected targets between snails of the ‘Saline group’, ‘ASA_Saline group’, ‘LPS group’, ‘Garcia effect group’, and ‘ASA_LPS group’.

A one-way ANOVA followed by Tukey’s post hoc test showed a main effect of the behavioral procedure on the expression levels of *LymTLR4* [F (4, 35) = 8.71, R^2^ = 0.49, *p* < 0.001] and *LymMDM* [F (4, 35) = 17.83, R^2^ = 0.67, *p* < 0.001] (Figure 3A,B). As observed in untrained animals, the LPS exposure (either in snails of the Garcia effect group or those of the LPS group) induced a significant upregulation of the mRNA levels of these targets compared to the non-LPS-injected counterparts (*LymTLR4*: LPS_C vs. Saline: *p* = 0.01, q = 4.86; LPS_C vs. ASA_Saline: *p* = 0.0005, q = 6.62; LPS_C vs. ASA_LPS group: *p* = 0.006, q = 5.28; Garcia effect vs. Saline: *p* = 0.03, q = 4.26; Garcia effect vs. ASA_Saline: *p* = 0.002, q = 5.26; Garcia effect vs. ASA_LPS group: *p* = 0.02, q = 4.68; *LymMDM*: LPS_C vs. Saline: *p* = 0.0001, q = 7.19; LPS_C vs. ASA_Saline: *p* < 0.0001, q = 7.69; LPS_C vs. ASA_LPS group: *p* = 0.0001, q = 7.65; Garcia effect vs. Saline: *p* < 0.0001, q = 7.49; Garcia effect vs. ASA_Saline: *p* < 0.0001, q = 8.09; Garcia effect vs. ASA_LPS group: *p* < 0.0001, q = 7.95).

Similarly, the expression levels of *LymHSP70* were upregulated [F (4, 35) = 13.03, R^2^ = 0.59, *p* < 0.001] in snails injected to LPS (i.e., snails the Garcia effect group and those of the LPS group) compared the other groups following the Garcia effect procedure (LPS_C vs. Saline: *p* = 0.0009, q = 6.19; LPS_C vs. ASA_Saline: *p* = 0.011, q = 4.91; LPS_C vs. ASA_LPS group: *p* = 0.004, q = 5.4; Garcia effect vs. Saline: *p* < 0.0001, q = 8.04; Garcia effect vs. ASA_Saline: *p* = 0.0003, q = 6.76; Garcia effect vs. ASA_LPS group: *p* < 0.0001, q = 7.26) (Figure 3C). No differences were found in the expression levels of *LymTLR4*, *LymMDM*, and *LymHSP70* between snails of the Garcia effect group and those of the LPS group (*LymTLR4*: *p* = 0.99, q = 0.59; *LymMDM*: *p* = 0.99, q = 0.29; *LymHSP70*: *p* = 0.68, q = 1.85).

Interestingly, a main effect of the Garcia effect formation was found on the expression levels of *LymGRIN1* [F (4, 35) = 16.38, R^2^ = 0.65, *p* < 0.001; Figure 3D], *LymGRIN2A* [F (4, 35) = 14.59, R^2^ = 0.62, *p* < 0.001; Figure 3E], *LymGRIN2B* [F (4, 35) = 7.69, R^2^ = 0.46, *p* < 0.001; Figure 3F] *LymGRIA1* [F (4, 35) = 15.57, R^2^ = 0.49, *p* < 0.001; Figure 3G], and *LymCREB1* [F (4, 35) = 14.27, R^2^ = 0.62, *p* < 0.001; Figure 3H].

In particular, Tukey’s multiple comparisons tests showed significant upregulation of the expression levels of these targets only in snails of the Garcia effect group compared to the other ones (*LymGRIN1*: vs. Saline: *p* < 0.0001, q = 7.05; vs. ASA_Sal: *p* < 0.0001, q = 9.08; vs. LPS_C: *p* < 0.0001, q = 7.27; vs. ASA_LPS: *p* < 0.0001, q = 10.46; *LymGRIN2A*: vs. Saline: *p* < 0.0001, q = 9.13; vs. ASA_Sal: *p* < 0.0001, q = 7.49; vs. LPS_C: *p* < 0.0001, q = 9.36; vs. ASA_LPS: *p* = 0.0006, q = 6.42; *LymGRIN2B*: vs. Saline: *p* = 0.003, q = 5.42; vs. ASA_Sal: *p* = 0004, q = 6.69; vs. LPS_C: *p* = 0.0007, q = 6.42; vs. ASA_LPS: *p* = 0.0013, q = 6.03; *LymGRIA1*: vs. Saline: *p* = 0.0002, q = 7.02; vs. ASA_Sal: *p* < 0.0001, q = 9.91; vs. LPS_C: *p* < 0.0001, q = 8.67; vs. ASA_LPS: *p* < 0.0001, q = 8.49; *LymCREB1*: vs. Saline: *p* < 0.0001, q = 7.37; vs. ASA_Sal: *p* < 0.0001, q = 9.60; vs. LPS_C: *p* < 0.0001, q = 6.81; vs. ASA_LPS: *p* < 0.0001, q = 8.59).

## 4. Discussion

Using *Lymnaea* as a model system, we examined using the Garcia effect procedure to explore novel aspects of this complex conditioned behavior and its transcriptional regulation. Building upon our previous findings, we successfully replicated the observation that exposure to a novel appetitive taste coupled with the LPS inducement of sickness followed by an LPS injection induces a specific feeding suppression known as the Garcia effect in freshly collected Margo snails. Interestingly, the dose of LPS used did not independently affect the snails’ feeding behavior or induce neophobia, yet it effectively triggered the Garcia effect. Moreover, our data confirmed previous studies demonstrating that ASA alone does not impact feeding behavior, but it efficiently prevents the effects mediated by LPS [27].

Given these promising results, we further investigated the transcriptional effects induced by the injection of 25 μg of LPS, which served as the sickness-inducing stimulus necessary for the formation of the Garcia effect. Our findings revealed that the LPS injection, both alone and in combination with the appetitive stimulus during the Garcia effect procedure, significantly increased the expression levels of *LymTLR4* and *LymMDM*. These genes are key mediators of the immune response [29,49]. Previous studies conducted in mammals, including humans, have demonstrated that LPS—through the stimulation of TLR4—induces the release of critical proinflammatory cytokines that are necessary to activate potent immune responses [31,50]. This immune activation, then, triggers sickness behavior [51,52], a well-characterized state that encompasses neuro vegetative and behavioral alterations.

Here, we also found that pre-exposure to ASA—an anti-inflammatory drug—before LPS injection, prevented this upregulation in *Lymnaea*. These findings suggest that inhibiting the activation of TLR4 by LPS prevents subsequent immune signaling in the CNS, suppressing the inflammatory cascade, the sickness state, and ultimately at the behavioral level, inhibiting the formation of the Garcia effect. Additionally, we observed a significant increase in *LymHSP70* expression when LPS was injected alone, before the presentation of carrot slurry, and when used as the sickness-inducing stimulus in the Garcia effect procedure. Heat shock proteins (HSPs), initially identified for their response to thermal stress, have been implicated in preventing protein misfolding, and recent evidence suggests their involvement in synaptic plasticity and memory formation [53,54]. In a previous study, we demonstrated that the upregulation of HSP70 induced by heat shock (i.e., used as nausea/sickness-inducing stimulus) is essential for Garcia effect memory formation when employing the carrot slurry-heat shock procedure [24]. The results presented in this study suggest that—similarly to the heath shock—the LPS-induced upregulation of HSP70 may play a key role in mediating the Garcia effect. Moreover, Porto et al. (2018) reported that in rats HSP70 is rapidly induced and modulates the MAPK-signaling pathway during memory consolidation in hippocampal neurons [55].

Interestingly, the orthologous gene of MAPK in *Lymnaea* (*LymMAPK*) and its related pathway are involved in connecting glutamate receptors to *LymCREB1* [37].

*LymCREB*-dependent genes are essential for modulating synaptic plasticity processes, including LTM [1,11]. Thus, our results show that an injection of LPS alone does not directly impact the expression levels of neuroplasticity genes such as *LymGRIN1*, *LymGRIN2A*, *LymGRIN2B*, *LymGRIA1*, and *LymCREB1*. However, when combined with the conditioning procedure for the Garcia effect, LPS leads to a significant upregulation of these genes which is associated with the formation of memory for the Garcia effect. These findings suggest that a molecular link between HSP70 and GRIN, GRIA, and CREB1 may exist in the nervous system of *Lymnaea,* which may play a role in mediating the Garcia effect. To our knowledge, this is the first evidence for the upregulation of the expression levels of *GRIA* and *GRIN 2A* and *2B* subunits in the central ring ganglia of pond snails, which formed LTM following a behavioral procedure.

Moreover, the results of this study suggest that there is a connection between targets involved in immune and stress responses and those that mediate learning and memory formation. The regulation of these targets, along with other signaling molecules, appears to be a coordinated effort that determines the memory phenotype. This synergistic regulation is essential for immune homeostasis, learning, and memory formation in *Lymnaea*, similar to what has been observed in mammals. The integration of immunological, neuronal, and stressful inputs in the central nervous system plays a crucial role in this regulation [49,56].

The ability to remember past experiences associated with aversive stimuli is important for survival and is conserved across different species, including humans [57,58]. Therefore, the results of this study open up possibilities for future research into the molecular cascades that underlie the complex interaction between immune stimulation and neuroplasticity. Understanding these conserved mechanisms could provide valuable insights into how organisms integrate immune responses and memory processes. The results of this study raise several questions. Firstly, the impact of higher doses of LPS on snails’ learning and memory formation remains unknown. It is unclear whether higher doses would completely hinder their ability to learn and form memories or if they would still associate the appetitive taste with the sickness induced by LPS. Further experiments will be conducted to address this question. Secondly, since *Lymnaea* has an open circulatory system, an injection of LPS can potentially affect not only the central nervous system but also the peripheral nervous system and other organs. In future experiments, we plan to compare the expression levels of selected targets across different tissues to gain a comprehensive understanding of their regulation. Thirdly, based on the findings of this study, we will investigate the involvement of other pathways in mediating immune responses and neuroplasticity in *Lymnaea*. Specifically, we will explore the roles of the endocannabinoid system and the kynurenine pathway, as they have been implicated in crosstalk between the immune and central nervous systems in mammals. Previous studies have highlighted their regulatory functions [59]. Finally, proteomic and metabolomic analyses will be conducted to examine the effects of the LPS-induced Garcia effect and its pharmacological regulation by ASA on homeostatic functions in *Lymnaea*, as well as their impact on neuroplasticity. These comprehensive analyses will provide insights into the broader physiological and molecular changes associated with the Garcia effect and its modulation by ASA.

## 5. Conclusions

In conclusion, our findings further validate the LPS-induced Garcia effect as a valuable learning paradigm for investigating the conserved molecular mechanisms underlying this form of learning and memory. Additionally, our study highlights the suitability of *Lymnaea* as an excellent model organism for studying both Neuroscience and Immunology. The ability of LPS to induce a Garcia effect in snails suggests the existence of a conserved communication network between the immune and central nervous systems. Furthermore, blocking LPS-induced sickness by ASA, a widely used anti-inflammatory drug, underscores the pivotal role of LPS in triggering an inflammatory response that alters behavioral adaptive responses. Here, we demonstrated that ASA, possibly the most popular drug of the modern era, is effective in preventing LPS-induced behavioral and molecular effects also in *Lymnaea*. ASA acts by inhibiting the synthesis of prostaglandins through acetylation of cyclooxygenases (COX) to reduce the inflammatory effect in the recipient, but also through COX-independent mechanisms, like the inhibition of nuclear factor (NF)-κB and the extracellular signal-regulated kinase (ERK) signaling. Our results support the prominent role of *Lymnaea* as a unique translational model and pave the way for its employment in the study of the molecular mechanism involved in neuro-immune pharmacology. Also, the results of this study may pave the way for future studies in mammals aimed at investigating the complex crosstalk between the immune and the central nervous system, as well as the conserved mechanisms underlying the Garcia effect.

By utilizing snail models, we can significantly reduce the reliance on mammalian models, limiting their involvement to result in validation and greatly reducing the costs associated with numerous studies. Finally, *L. stagnalis* as a model system provides an important experimental tool and offers a translational approach that contributes significant insights and understanding in the field of Neuroscience, Immunology, and Pharmacology.

## Figures and Tables

**Figure 1 biology-12-01100-f001:**
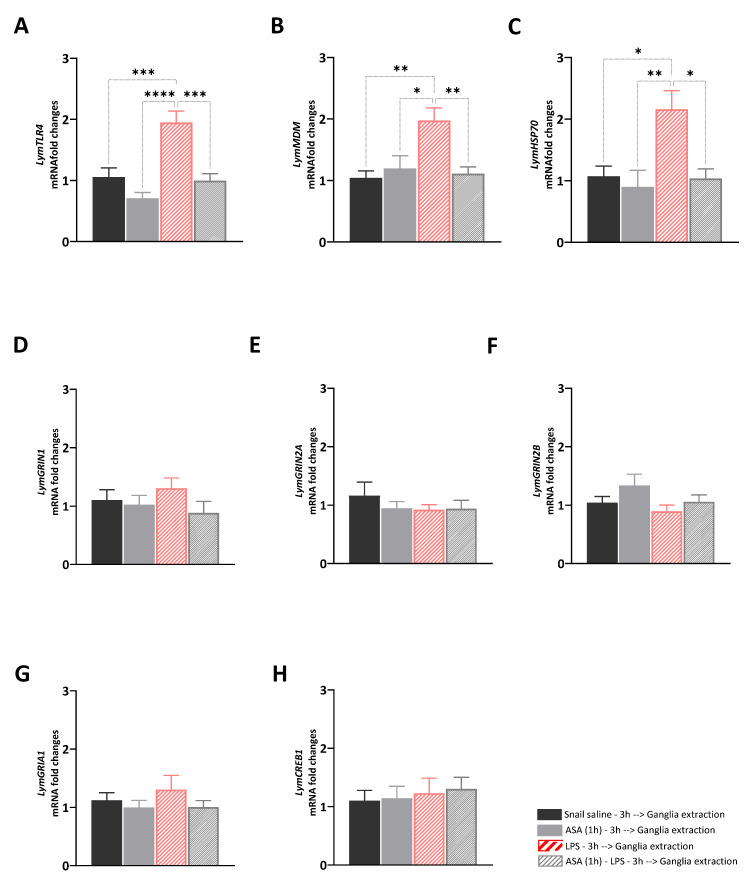
Transcriptional effects induced by ASA treatment, LPS injection, and their paired presentation. The expression of *LymTLR4* (**A**), *LymMDM* (**B**), *LymHSP70* (**C**), *LymGRIN1* (**D**), *LymGRIN2A* (**E**), *LymGRIN2B* (**F**), *LymGRIA1* (**G**), and *LymCREB1* (**H**) were measured in the central ring ganglia of snails injected with snail saline (full black bars), snails exposed to ASA for 1 h (full grey bars), snails injected with LPS (diagonal red bars), and snails exposed to ASA for 1 h and then injected with LPS (diagonal grey bars). Three hours after the treatment, snails were sacrificed, and the central ring ganglia were extracted. The mRNA levels were assessed using RT-qPCR. The sample size (N) for each group was 8. The data are presented as means ± SEM and were subjected to statistical analysis using One-way ANOVA, followed by Tukey post hoc analyses. Statistical significance was indicated as **** *p* < 0.0001, *** *p* < 0.001, ** *p* < 0.01, * *p* < 0.05.

**Figure 2 biology-12-01100-f002:**
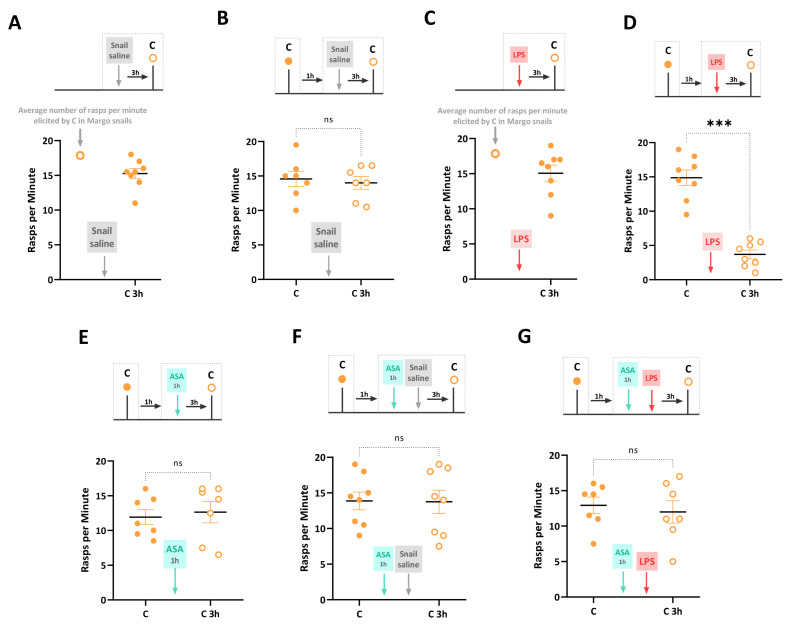
LPS-induced Garcia effect and its pharmacological regulation by ASA. The experimental timeline and results are summarized as follows: (**A**) The Saline_C group, consisting of 8 naïve Margo snails, was injected with snail saline. After 3 h, snails were exposed to carrot slurry for the first time (closed circles). The injection of snail saline did not affect the feeding behavior elicited by carrot slurry, as the number of rasps per minute was not significantly different from that observed in non-injected snails. (**B**) A control group of 8 freshly collected Margo snails was exposed to carrot slurry 1 h before (closed circles) and 3 h after (open circles) the snail saline injection. There were no significant differences in the number of rasps elicited by carrot slurry between the two-time points. (**C**) A naïve cohort of snails (LPS group, N = 8) was first injected with LPS, and after 3 h, the rasping behavior in response to the carrot slurry was counted. The injection of LPS before the snails ever experienced the carrot slurry did not alter the response to the taste when compared to control non-injected snails. (**D**) The Garcia effect was observed in snails (Garcia effect group, N = 8) exposed to carrot slurry 1 h before (closed circles) and 3 h after (open circles) the LPS injection. The exposure to the novel taste (carrot slurry) followed by an LPS injection resulted in a significant taste-specific reduction in the number of rasps, indicating the formation of the Garcia effect. (**E**) In a new cohort of 8 naïve snails (ASA group), the number of rasps elicited by carrot slurry was counted 1 h before (closed circles) and 3 h after (open circles) a 1 h exposure to ASA. There were no significant differences in the number of rasps, indicating that ASA did not significantly alter the positive hedonic effect of carrot slurry (**F**) The ASA_Saline group (N = 8) was exposed to carrot slurry for 2 min (closed circles). One hour later, they were exposed to ASA for 1 h, immediately followed by an injection of snail saline. Three hours later, the number of rasps in the carrot slurry was re-counted (open circles). The paired treatment of ASA and saline injection did not significantly affect the feeding behavior elicited by carrot slurry. (**G**) The ASA_LPS group, comprising naïve snails (N = 8), was exposed to carrot slurry, and immersed in ASA for 1 h before being injected with LPS. The number of rasps was counted at 3 h post-injection (open circles). No significant differences in the number of rasps elicited by carrot slurry were observed. The data are presented as means ± SEM and were analyzed using paired *t*-tests (**B**,**E**–**G**) or unpaired *t*-tests (**A**,**C**). The statistical significance level was represented as *** for *p* < 0.001, and ns indicated no significance with *p* > 0.05.

**Figure 3 biology-12-01100-f003:**
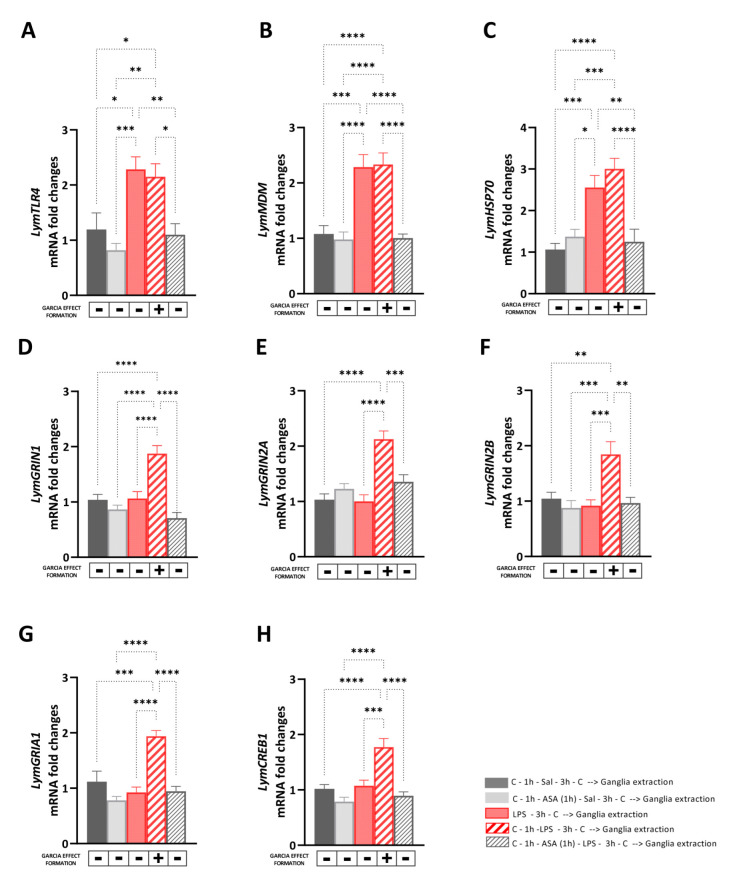
Transcriptional effects induced by the LPS-induced Garcia effect and its pharmacological regulation mediated by ASA. The expression of *LymTLR4* (**A**), *LymMDM* (**B**), *LymHSP70* (**C**), *LymGRIN1* (**D**), *LymGRIN2A* (**E**), *LymGRIN2B* (**F**), *LymGRIA1* (**G**), and *LymCREB1* (**H**) was measured in the central ring ganglia of (1) snail of the ‘Saline group’ (full black bars) were exposed to carrot slurry 1 h before and 3 h after being injected with snail saline; (2) snails of the ‘ASA_Sal group’ (full grey bars), which were first exposed to the carrot slurry 1 h before and 3 h after being exposed to ASA for 1 h and immediately later injected with snail saline; (3) snails of the ‘LPS group (full red bars)’, which were first injected with LPS and three hours later, and were exposed for the first time to carrot slurry; (4) snails of the ‘Garcia effect group’ (diagonal red bars), which were exposed to carrot slurry 1 h before and 3 h after being injected with LPS; and (5) snails of the ‘ASA_LPS group’ (diagonal grey bars), which were exposed to carrot slurry 1 h before and 3 h after being exposed to ASA for 1 h and immediately after were injected with LPS. Snails that learned and formed the Garcia effect memory are indicated with a ‘+’ below each bar, whereas those which did not form the Garcia effect were marked with a ‘-’ below the bars. After the exposure to the carrot slurry, snails were sacrificed, the central ring ganglia were dissected, and the RNA was extracted. RT-qPCR was employed to analyze the mRNA levels in the study, with a sample size of 8 for each group. The data were presented as means ± SEM and statistically analyzed using One-way ANOVA, followed by Tukey post hoc analyses. Statistical significance was denoted as **** *p* < 0.0001, *** *p* < 0.001, ** *p* < 0.01, * *p* < 0.05.

**Table 1 biology-12-01100-t001:** The nucleotide sequences of the forward and reverse primers utilized for RT-qPCR are provided. Additionally, the accession number and size (in base pairs) of the PCR product obtained through cDNA (mRNA) amplification for each target are also specified.

Gene Bank Accession	Target	Product Length (bp)	Type Sequence
X15542.1	*Snail*, *beta-tubulin* *LymbTUB*	100 bp (92–192)	FW: GAAATAGCACCGCCATCC
			RV: CGCCTCTGTGAACTCCATCT
DQ278441.1	*Lymnaea stagnalis elongation factor 1-alpha*, *LymEF1α*	150 bp (7–157)	FW: GTGTAAGCAGCCCTCGAACT
			RV: TTCGCTCATCAATACCACCA
AY577328.1	*Lymnaea stagnalis Toll-like receptor 4* *LymTLR4*	100 bp (74–174)	FW: GGAGGGTCAAGCATAAAGTGT
			RV: CATCAAGGTCAACGCCAAT
U58769.1	*Lymnaea stagnalis molluscan defense molecule precursor* *LymMDM*	104 bp (1614–1718)	FW: CGGGTACACACACAGATGGA
			RV: TGACTGAACATTGGGCACAC
DQ206432.1	*Lymnaea stagnalis heat-shock protein 70* *LymHSP70*	199 bp (134–333)	FW: AGGCAGAGATTGGCAGGAT
			RV: CCATTTCATTGTGTCGTTGC
AY571900.1	*Lymnaea stagnalis NMDA-type glutamate receptor subunit 1* *LymGRIN1*	140 bp (831–917)	FW: AGAGGATGCATCTACAATTT
			RV: CCATTTACTAGGTGAACTCC
FX180835	*Lymnaea stagnalis NMDA-type glutamate receptor subunit 2A* *LymGRIN2A*	129 bp (3454–3583)	FW: GATCACCAAGGATGATTACT
			RV: CTTGGCTATATTCAAGTCTGT
FX180839	*Lymnaea stagnalis NMDA-type glutamate receptor subunit 2B* *LymGRIN2B*	126 bp (4147–4273)	FW: GACTCCTCTGTTTTGGAATA
			RV: GGTTCCTTGATGGTTTATTA
FX183516.1	*Lymnaea stagnalis AMPA-type glutamate receptor subunit 1* *LymGRIA1*	111 bp (1205–1316)	FW: AGACTGTTGTAGCTGTCCTT
			RV: ATAGCTATTGGATTTCTTGC
AB041522.1	*Lymnaea stagnalis cAMP responsive element binding protein* *LymCREB1*	180 bp (49–229)	FW: GTCAGCAGGGAATGGTCCTG
			RV: ACCGCAGCAACCCTAACAA

## Data Availability

The data presented in this study are available in the present article and on request from the corresponding author.

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
