# Peer review of "LPS-Induced Garcia Effect and Its Pharmacological Regulation Mediated by Acetylsalicylic Acid: Behavioral and Transcriptional Evidence"

_biology, 2023, doi:10.3390/biology12081100_

Round 1

Reviewer 1 Report

The manuscript entitled „LPS-induced Garcia effect and its pharmacological regulation mediated 2 by acetylsalicylic acid: behavioral and transcriptional evidence” is, in my opinion, interesting and of great significance. The authors were the first to demonstrate the existence of the Garcia effect in invertebrates and have made a great effort so far to reveal the aspects of this learning paradigm in Lymnaea which has been used as a model in invertebrate neuroscience for decades. According to this, the manuscript builds on the Authors’ previous findings. Since, to the best of my knowledge, the Garcia effect was previously investigated only at the behavioural level in Lymnaea, the Authors’ initiative to understand the underlying cellular and molecular mechanisms are welcome. However, I have some concern regarding the manuscript. I believe that the Authors will able to address my issues.

- At some places in the text the injected LPS is written as 25 mg but at some places it is 25 µg. I suppose it was 25 µg. Please unify it it in the text

- Introduction – Line 66-77: I strongly feel that this paragraph is not neccesary in the Introduction. This manuscript deals with the LPS-induced Garcia effect but the paragraph present previous findings regarding the heat shock-induced Garcia effect. I highly appreciate the Authors’ previous works with the heat shock-induced Garcia effect, especially the comparison of lab-bred and field-collected snails, but I think that presenting these findings does not add any relevant information to the content of the manuscript. Please consider removing it.

- Methods: please check the description of Experiment 2, it does not match with Fig 2. For example, the description of of ’LPS-C’ group and ’LPS group’ are actually the same.

- Results:

1) My major concern with the manuscript is that the presented results in Figure 2, expect maybe Fig2A, were previously published in the Rivi et al. Novel taste, sickness, and memory: Lipopolysaccharide to induce a Garcia-like effect in inbred and wild strains of Lymnaea stagnalis. Physiol Behav. 2023 263:114137 paper. I understand that the gene expression measurements are based on these behavioural results and I am very happy that the previously published results were consistent and reproducible (sadly I see a lot of published papers where I doubt the reproducibility) but I strongly feel that, given these findings are already published, Figure 2 should be placed in the Supplementary information.

2) How can the Authors explain/interpret that ASA alone cannot induce any gene expression changes but it can prevent the LPS-induced gene expression changes?

3) How did the Authors identify the Lymnaea NMDA subunits? If I remember well, there is only a few information on this in Lymnaea and even in Aplysia. NMDA1 (GluN1, GRIN1) subunit is definitely O.K., Moroz and his co-workers identified it many years ago. But, if I know well, the other subunit(s) has/have never been identified with 100% certainty and there is no information on the potential receptor structure. There are also unanswered questions about the function since there is no Mg2+ block. The analysis of the sequences included by the Authors in Table 1 does not indicate that they would be NDMA receptor subunits.

- The future plans presented in the end of the Discussion section are promising. I suggest the Authors check the Seppälä et al. Transcriptome profiling of Lymnaea stagnalis (Gastropoda) for ecoimmunological research. BMC Genomics. 2021 22(1):144. paper where a lot of immune defense factors including cytokines have been identified.

Author Response

We would like to thank Reviewer 1 for the time and effort that she/he dedicated to providing feedback on our manuscript and are grateful for the insightful comments on and valuable improvements to our paper.

In the attached file, we provided a point-by-point response to her/his comments and believe that the changes made (marked in red within the manuscript) have improved the quality of our paper. All line numbers refer to the revised manuscript.

We hope the Reviewer will now judge the revised manuscript as fit for publication in the Biology Journal.

Reviewer 2 Report

Rivi et al. investigated transcriptional changes in the CNS of the pond snail Lymnaea following the formation of Garcia effect memory. They found that several immune response-related genes are up-regulated by LPS injection, and some neuronal plasticity-related genes are up-regulated only when LPS injection was paired with a novel food odor. They also found ASA treatment, which suppresses the memory formation, inhibits the transcriptional changes caused by LPS and Garcia effect memory formation. These data demonstrate the parallelism between transcriptional changes and memory formation, as well as immune response in the CNS. Overall, the data of qRT-PCR and behavioral study are sound and clear. But the histochemical data showing the locus of plastic changes in the CNS would be helpful.

Major comments

Line 365-374, 375-381

Statistical comparison between Saline group and Saline_C group appears to be duplicated. Please read carefully these paragraphs.

Line 458-462

ASA exerts its anti-inflammatory effects by inhibiting COX, whose expression/activity is regulated under the signaling of TLR4. Therefore, it is reasonable to think that the inflammation caused by LPS-induced activation of TLR4 can be suppressed by ASA. But how does ASA treatment inhibit up-regulation of TLR4 mRNA? If the authors have information regarding such a reverse regulatory relationship, please explain. In relation, do the authors have any biochemical evidence for the effectiveness of ASA in inhibition of COX in Lymnaea?

Line 109-113, 476-486

The authors focused on LymGRIN1, 2A, 2B, LymGRIA1 because these ion channels are involved in neuroplasticity (line 109-113). Are there any other reports on the up-regulation of these genes in other learning paradigms or in synaptic plasticity of Lymnaea? In mammals, for example, the protein of AMPA receptor increases during long-lasting hippocampal LTP (Nayak et al., Nature 394, 680-3, 1998). If the authors’ finding, i.e. learning-dependent up-regulation of ion channel genes, is completely novel in Lymnaea, this point should be stressed. The reviewer is also curious about the locus of induction of these genes in the CNS of Lymnaea. Do the authors have any data suggesting the loci where plastic changes occur during Garcia effect memory formation?

Minor comments

Line 35

Spell out LPS here, not only in line 78.

Line 79-83

Why the periods of memory retention are different between lab-bred strain and pond-collected snails? Pease provide a brief explanation.

Line 107

LymCREB1 should be spelled out here, not in Line 111.

Line 120-121

English is somewhat strange.

Line 148

Describe the manufacturer of aspirin.

Line 232

elongation factor 1apha and beta-tubulin, not simply tubulin.

Line 363

“retrotranscribed” sounds a little bit strange. Shouldn’t it be “reverse-transcribed”?

Figure 1

The pattern of the rectangle of “ASA(1h)-3h-->ganglia extraction” does not match those in the graphs. The former is hatched, but the latter is light gray.

Figure 2

Letters in Fig. 2A, C are very small and hard to read. Also, please provide a reason why only the average number of rasps are shown in these data.

Figure 3

The letters in the left below of each graph (A-H) are very small.

Author Response

We would like to thank Reviewer 2 for the time and effort that she/he dedicated to providing feedback on our manuscript and are grateful for the insightful comments on and valuable improvements to our paper.

In the attached file, we provided a point-by-point response to her/his comments and believe that the changes made (marked in red within the manuscript) have improved the quality of our paper. All line numbers refer to the revised manuscript.

We hope the Reviewer will now judge the revised manuscript as fit for publication in the Biology Journal.

Round 2

Reviewer 1 Report

The authors answered my questions well and made the necessary revisions. Great job!

I have just one minor but important comment: The ref list in the revised version seems to be messed a bit. For example, in the original version, ref 1-10 were about the Lymnaea learning/neuro model in general and ref 11-16 were about the relevant papers about the Garcia effect but in the revised version these refs (with other refs such as the Juhasz et al 2022 paper) seem to be mixed somehow. For example, ref 1,7,8,9,10 in the revised version are not relevant to the first sentence. Many previous relevant citations (e.g., Fodor 2020; Ito 2013; Kemenes 1989 papers) disappeared. 

Please carefully check the citations thought the manuscript. 

Author Response

We would like to thank Reviewer 1 for the time and effort that she/he dedicated to providing feedback on our manuscript and are grateful for the insightful comments on and valuable improvements to our paper.

We are terribly sorry, but there has been a problem with the software we used for creating the bibliography (i.e., Zotero). We thank Reviewer 1 for pointing this out. We have now included the missing references and adjusted the wrong ones. 

We hope the Reviewer will now judge the revised manuscript as fit for publication in the Biology Journal.

Reviewer 2 Report

The ms has been improved according to the reviewer's suggestion, except for one point: Letters in Figure 2A, C are still very small to read. They seem not to be enlarged in the revised ms at all. The authors should amend this point before final acceptance for publication. 

Author Response

We would like to thank Reviewer 2 for the time and effort that she/he dedicated to providing feedback on our manuscript and are grateful for the insightful comments on and valuable improvements to our paper.

As required by the Reviewer, we have now edited Figure 2A and Figure 2C increasing the Font size. We are confident that in the published version of our manuscript, the readers will be allowed to zoom the figures and easily read the content.

We hope the Reviewer will now judge the revised manuscript as fit for publication in the Biology Journal.